# Global Regulatory Frameworks for the Use of Artificial Intelligence (AI) in the Healthcare Services Sector

**DOI:** 10.3390/healthcare12050562

**Published:** 2024-02-28

**Authors:** Kavitha Palaniappan, Elaine Yan Ting Lin, Silke Vogel

**Affiliations:** Centre of Regulatory Excellence, Duke-NUS Medical School, Singapore 169857, Singapore

**Keywords:** regulatory frameworks, artificial intelligence, healthcare services

## Abstract

The healthcare sector is faced with challenges due to a shrinking healthcare workforce and a rise in chronic diseases that are worsening with demographic and epidemiological shifts. Digital health interventions that include artificial intelligence (AI) are being identified as some of the potential solutions to these challenges. The ultimate aim of these AI systems is to improve the patient’s health outcomes and satisfaction, the overall population’s health, and the well-being of healthcare professionals. The applications of AI in healthcare services are vast and are expected to assist, automate, and augment several healthcare services. Like any other emerging innovation, AI in healthcare also comes with its own risks and requires regulatory controls. A review of the literature was undertaken to study the existing regulatory landscape for AI in the healthcare services sector in developed nations. In the global regulatory landscape, most of the regulations for AI revolve around Software as a Medical Device (SaMD) and are regulated under digital health products. However, it is necessary to note that the current regulations may not suffice as AI-based technologies are capable of working autonomously, adapting their algorithms, and improving their performance over time based on the new real-world data that they have encountered. Hence, a global regulatory convergence for AI in healthcare, similar to the voluntary AI code of conduct that is being developed by the US-EU Trade and Technology Council, would be beneficial to all nations, be it developing or developed.

## 1. Introduction

The healthcare sector is currently facing several challenges that are not localised but rather universal, such as manpower shortages and difficulties in chronic disease management [1]. The demand for and the size of the global health workforce are predicted to rise to unprecedented levels because of population and economic growth [2]. Simultaneously, chronic, non-communicable diseases (NCDs) are reported to be the leading cause of global death, at about 73%, with millions of individuals being continuously affected [3]. The current shortage of labour in healthcare professionals is further exacerbated by demographic and epidemiological shifts. The ageing population and rise in NCDs in high-income countries (HICs), as well as low- and middle-income countries (LMICs), means that healthcare delivery models have to expand to meet population needs, especially in the areas of patient services and community care [2]. Nations around the world have started to realise that health systems need to be built upon foundational models of care that encompass preventive and rehabilitative services in conjunction with the tremendous potential of digital technologies to meet the rising population demands [2].

Digital health interventions that include artificial intelligence (AI) have been recognised to be capable of helping to fill these gaps in healthcare [4]. According to the Encyclopaedia Britannica, AI is defined as “the ability of a digital computer or computer-controlled robot to perform tasks commonly associated with intelligent beings” [5]. While AI is computer software that mimics the ways that humans think in order to perform complex tasks, such as analysing, reasoning, and learning, machine learning (ML) on the other hand is a subset of AI that uses algorithms trained on data to produce models that can perform such complex tasks [6]. The rising attention and surge in AI applications are attributed to the synergy between the significant advancement of computational powers and the massive volume of data generated from health systems [7]. AI is increasingly playing a significant role in redefining and revolutionising the existing healthcare landscape, from the automation of administrative functions supporting diagnosis through evidence-based clinical decision making to suggesting suitable treatments by analysing massive quantities of health data at a rapid speed [8]. The application of AI in healthcare has been shown to improve patient health outcomes and the well-being of healthcare professionals [9].

At the same time, it is necessary to note that AI-based technologies are still in a nascent phase and hence require various actors, from AI researchers and developers to regulatory authorities, to be informed on the developments of AI. Out of all these actors, regulatory authorities play a crucial role as AI systems in healthcare make critical decisions that directly impact individuals’ health, safety, and well-being, and regulations help to prevent errors or malfunctions that could potentially harm patients. Additionally, regulations provide a framework for the ethical use of AI in healthcare. They address issues such as patient privacy, consent, and the responsible handling of sensitive medical data. Without proper regulations, there is a risk of misuse or unauthorised access to personal health information.

Standardisation is another key aspect. Regulations help to establish common guidelines and standards for AI applications in healthcare, promoting interoperability and compatibility among different systems, which is essential for seamless collaboration and communication within the healthcare ecosystem. Furthermore, regulations can foster trust among healthcare professionals, patients, and the public. When people are aware that the AI systems adhere to established standards, it can increase confidence in their reliability and effectiveness.

## 2. Role of AI in Healthcare Services

Healthcare services are rapidly expanding to include remote and mobile modes of delivery, and hence, the incorporation of AI technologies to aid in the diagnosis, treatment, and prevention is timely and crucial. Currently, the use of AI in healthcare services ranges from prevention, diagnosis, and pharmacology to treatment. It is crucial to understand its impact on healthcare services for effective healthcare delivery across the healthcare continuum [10]. AI is mainly used for assisting and automating the existing healthcare services as far as the diagnostics and pharmaceutical areas are concerned, and when it comes to treatment aspects, it aims to augment the current healthcare services, as shown in Figure 1.

### 2.1. AI for Diagnosis and Prevention

In the area of diagnosis, studies have shown that AI algorithms perform on the same level or even better than clinicians, which may be partly due to the ability of AI algorithms to achieve high speed and accuracy in data interpretation [11]. Currently, AI in diagnostics is mainly used to read images and support clinicians in their decision-making process. For example, in a study that involved more than 112,000 images of chest X-ray scans used for the detection of pneumonia, AI algorithms showed better performance than radiologists [11]. In another study that involved the optical diagnosis of polyps at high magnification during colonoscopy with 325 patients, the speed of optical diagnosis with the AI algorithm was 35 s, with 94% accuracy [12,13].

Currently, certain AI applications (apps) and devices are also directly available to consumers for their personal use; these are known as direct-to-consumer (DTC) medical AI/ML apps [14]. The majority of these apps are for prevention purposes, and consumers use them to monitor their health status and prevent diseases before they can occur. As per companies’ own validation, their predictive functions are mostly accurate. Some of the apps have already been authorised by the United States (US) Food and Drug Administration (FDA) to provide a screening decision independent of a healthcare professional [15]. Examples include the ECG app in the Apple Watch, which is marketed and used by consumers to enable the personal screening of certain heart disorders, in particular atrial fibrillation [16].

Another example of the effective use of AI in preventive care involves healthcare services that may include AI for personalised nutrition for the management of chronic diseases well before their occurrence. There is also the possibility of patients managing their own medical conditions, especially chronic conditions such as diabetes, hypertension, and mental health issues with the help of AI [11].

### 2.2. AI in Pharmacology

In pharmaceutical research, AI is used for drug discovery and the prediction of chemical and pharmaceutical properties [17]. For instance, the drug synthesis process within the research and development (R&D) cycle could be shortened by using ML models to automate chemical experiments, as these models are able to perform thousands of chemical reactions simultaneously [18]. This will thus enable cost savings for conducting experiments and help researchers to offload repeated work. AI, ML, and robotics are currently paired with high-throughput screening and high-throughput experimentation to develop new drugs for specific patients in need [19].

Digital twins are gaining traction for application in drug discovery and efficacy studies. Digital twins are virtual representations of a particular object, and in healthcare research, a digital twin can be created for a person’s specific organ such as heart, liver, and kidney, or the entire person itself. The digital twin would be an exact replica down to the cellular level. The bioactivity, chemical, and pharmacological properties of new drugs are studied in such digital twins instead of actual organs [20].

### 2.3. AI for Treatment

AI is being integrated into treatment for patients. This includes the use of AI to optimise the drug delivery system to make treatment more effective. A micro- or nanosensor programmed with an AI algorithm is used to detect subtle changes in vivo and monitor drug concentrations, thereby generating feedback systems [21]. Such feedback systems can in turn be used for training the AI algorithm further, and once the AI algorithm is fully trained, the delivery system can facilitate self-medication. This process enables patients to adjust their drug dosage according to objective measures and, at the same time, also allows them to transfer their data to their physician via the cloud in real time [22]. Physicians would also be able to monitor the changes and suggest modifications if required via telemedicine. This would not only make treatment more precise, effective, and time-sensitive but also solve the issues related to the manpower crunch that the healthcare system is currently facing [23].

AI technologies are able to assist clinicians in evidence-based decision making with high accuracy and speed and perform on the same level or even outperform clinicians. Clinicians are currently using AI in a computer-aided system to better diagnose and treat breast cancer and lung cancer [24].

The generation of treatment plans is another area where AI is currently being explored. AI can generate treatment plans within a few minutes or even seconds, in comparison to the duration taken by physical providers [25].

AI is valuable in supporting the increasing resource demands in healthcare services, especially due to the ageing population. AI technologies may be deployed at nursing homes and patients’ residences to assist them in monitoring their conditions regularly in the form of robotic companions [26]. Such robotic companions may go beyond the scope of a medical device; they can even be interactive to provide advice to the elderly about their physical activities in their living space [27].

The shift towards home-based care from hospitals has largely been enabled by AI technologies such as telemedicine [23]. Telemedicine refers to the delivery of healthcare services at a distance by healthcare professionals using information and communication technologies to diagnose, treat, and prevent diseases and injuries [23]. Remote monitoring systems such as video-observed therapy and patient care made possible by virtual assistants are commonly seen as part of this shift [25]. Clare&Me is an AI virtual chatbot that offers support and guidance for mental health in Germany in a conversational manner via phone calls and WhatsApp [28]. Ora is a telehealth platform based in Singapore with an emphasis on healthcare delivery for the Southeast Asian population through a direct-to-patient model on issues pertaining to men’s health, female reproductive healthcare, and skin care treatment [29].

Several technology companies, including Alibaba, Tencent, Baidu, and Ping An, are now getting increasingly involved in healthcare delivery by starting to offer healthcare services directly to healthcare facilities and individuals [30,31,32,33]. While Alibaba is developing AI-assisted diagnostic tools [30], Tencent is creating a “smart hospital” that provides online services [31]. Baidu Health offers consultation and chronic disease management services, whereas Ping An built an AI technology called AskBob in 2019, which is an AI-based diagnosis and treatment assistance tool for doctors [34,35]. DispatchHealth is another company that serves as an urgent call centre on wheels and sends over help to online requesters depending on the medical services needed, including on-site blood tests [36]. DearDoc is yet another company that uses AI and automation to perform extensive research for diagnostics [37]. Such companies may not come under the direct purview of the regulation authorities of the healthcare system and hence may need to be regulated, especially when they offer direct provisions of healthcare services to the public.

Another emerging concept is the “uberization” of healthcare. In this concept, AI creates common healthcare platforms where various healthcare professionals can acquire work on demand [38].

The economic perspective of the use of AI in healthcare seems to be hypothetically positive. A study examining roughly 200 studies for AI in healthcare revealed that there were tremendous cost savings using AI tools in both diagnosis and treatment when compared to conventional methods. Cost savings in diagnosis in the range of USD 1666 per day per hospital in the first year to about USD 17,881 per day per hospital in the tenth year and cost savings in treatment in the range of USD 21,666.67 per day per hospital in the first year to about USD 289,634.83 per day per hospital in the tenth year were observed [39]. Likewise, another study has also shown that cost savings due to the use of AI in treatment are much more than cost savings due to the use of AI in diagnosis [40].

Thus, based on the increased use of AI in various healthcare settings, as described above, our goal is to examine the scope of the existing regulatory landscape in encompassing the various risks associated with AI and its use in the healthcare sector.

## 3. Methodology

A review of the existing literature was conducted to scan the horizon for the regulations pertaining to AI in the healthcare sector. Based on a recent study conducted, seven jurisdictions were identified to be pioneers in the area of regulating AI in healthcare, namely the United States of America (USA), the United Kingdom (UK), Europe, Australia, China, Brazil, and Singapore [41]. The regulatory frameworks and guidelines pertaining to AI in healthcare in these seven jurisdictions were identified and downloaded from their respective government websites pertaining to healthcare and analysed for the purpose of this review. The key terms used for the search included regulatory frameworks, legislations, laws, policies, and guidelines.

In terms of the policies presented in this review, a mixture of hard and soft laws was considered. Hard laws refer to legislation or a set of rules that are legally binding [42]. Soft law comprises professional guidelines, voluntary standards, codes of conduct, recommendations, agreements, national action plans, or policy documents, which are not legally binding and adopted by governments and the industry [43]. The review also includes national policies that are in draft or implementation stages developed by the government, their agencies, and national standard bodies.

## 4. Existing Regulatory Landscape

Across the global regulatory landscape, the use of AI in healthcare is currently predominantly regulated under the regulatory frameworks for medical devices, or more specifically, under the frameworks of Software as a Medical Device (SaMD). Several articles have addressed the details, pros and cons of such regulations [44]. It is important to note that these regulations do not apply to certain AI applications such as software intended to support people in maintaining a healthy lifestyle, software used for administrative support, and software that provides clinical support or recommendations to healthcare professionals. This is mainly because the individual is expected to be qualified to make his or her own rational decisions based on the recommendations provided by the AI application [45].

The analysis of global regulatory frameworks for the use of AI in healthcare shows that regulations currently mostly adopt a soft-law approach. Examples of these soft-law approaches include professional guidelines, voluntary standards, and codes of conduct that are adopted by governments and the industry [43]. For AI in healthcare, considering soft-law frameworks, there are substantial expectations for the relevant stakeholders to consider during the development of AI’s technological innovations; however, they are not directly enforced by governments [46]. The relevant stakeholders, in this case, refer to developers and users. The term “developer” refers to a singular person or an organisation that is involved in the planning, funding, developing, and/or maintaining AI-MDs; the term “user” refers to a singular person or an organisation that uses AI-MDs in the delivery of healthcare services [47].

The benefits of adopting a soft-law approach are that it can be easily amended given the evolving landscape of AI technologies. The downside is that these approaches are voluntary; thus, organisations have the option not to adopt these voluntary guidelines.

### 4.1. United States of America (USA)

Currently, there are no specific regulatory pathways for AI-based technologies in the USA, but the FDA evaluates them under the existing regulatory framework for medical devices [48]. In April 2019, the FDA implemented the “Proposed Regulatory Framework for Modifications to AI/ML-based SaMD” according to which developers were accountable for the real-world performance of their AI systems and needed to update the FDA on the changes in terms of performance and input [49]. The proposal also emphasised that the approval process needs to restart if there is a change in the intended use of the AI system [50]. Following this proposal, the FDA issued the “AI/ML-based SaMD Action Plan” in January 2021, which outlined the following five actions based on the total product life cycle (TPLC) approach for the oversight of AI-MDs [51,52]:Specific regulatory framework with the issuance of draft guidance on “Predetermined Change Control Plan”;Good machine learning practices;Patient-centric approach, including the transparency of devices to users;Methods for the elimination of ML algorithm bias and algorithm improvement;Real-world performance monitoring pilots.

The guidance to the “Predetermined Change Control Plan” is expected to be a framework for modification to AI-MDs and would include the type of anticipated modifications, known as “SaMD pre-specifications” (SPSs), and the associated methods used to implement those changes in a controlled manner that would mitigate the risks to patients, known as the “algorithm change protocol” (ACP) [52]. Good machine learning practices (GMLPs) were also included as part of the TPLC approach for AI-MD developers. GMLP considerations for SaMDs refer to good software engineering practices or quality system practices that include the following features:High relevance of available data to the clinical problem and current clinical practice;Consistency in data collection that does not deviate from the SaMD’s intended use;Planned modification pathway;Appropriate boundaries in the datasets used for training, tuning, and testing the AI algorithms;Transparency of the AI algorithms and their output for users [50].

### 4.2. United Kingdom (UK)

UK’s National Institution for Health and Care Excellence (NICE) collaborated with the National Health Service (NHS) England and published the “Evidence Standards Framework for Digital Health Technologies” in 2019. This document provides the regulations for a range of products such as apps, software, and online platforms that can be standalone or combined with other health products [53].

In addition, The Regulatory Horizons Council of the UK, which provides expert advice to the UK government on technological innovation, published “The Regulation of AI as a Medical Device” in November 2022 [54]. This document considers the whole product lifecycle of AI-MDs and aims to increase the involvement of patients and the public, thereby improving the clarity of communication between regulators, manufacturers, and users.

In September 2021, the Medicines and Healthcare Products Regulatory Agency (MHRA) established a regulatory reform programme known as the “Software and AI as a Medical Device Change Programme” to provide a robust regulatory framework in the form of guidance for the regulatory oversight of AI-MDs. The programme comprises two workstreams: the first stream considers key reforms across the whole lifecycle of SaMDs, which includes cybersecurity and data privacy risks, and a post-market evaluation of the medical device; the second considers additional challenges that AI can pose to medical device regulation, including evolving AI algorithms, bias, and the interpretability of AI [54].

### 4.3. Europe

The European Union (EU) began to establish its approach to AI with non-binding guidelines, including the “Ethics Guidelines for Trustworthy AI” [55] and the “Policy and Investment Recommendations”, which were published in 2019 [56]. Subsequently, in May 2021, the EU took a regulatory stand by publishing the “European Medical Device Regulation”, wherein the risk classification of SaMDs was based on diagnostic and therapeutic intentions. However, in April 2021, the EU proposed the AI Act, which laid down a harmonised legal framework for AI products and services, from the development phase to their application [57]. In that framework, Articles 9 to 15 address the requirements for AI systems with respect to risk management, data governance, human oversight, transparency, accuracy, robustness, and cybersecurity. In addition, the obligations of providers to users of such AI systems are provided in Articles 16 to 29. Thus, it is evident that the EU is currently moving from a soft-law approach towards a legislative approach for the regulatory framework in AI [58].

The AI Act uses a risk-based approach to regulate AI systems. In the healthcare sector, high-risk AI systems include those that utilise biometric identification, sort patients based on their medical history, and use software for the management of public healthcare services and electronic health records [59]. The main requirements for these high-risk AI systems under the AI Act are data governance and risk management, which need to be addressed by the manufacturer. For low- and minimal-risk AI systems such as chatbots that may interact with humans as part of healthcare service, a voluntary code of conduct for safe and reliable service needs to be in place [60]. Critics indicate that the AI Act is inflexible, as there is currently no scope to include new AI applications in the “high-risk” category if they emerge in an unforeseen sector and are dangerous [61].

### 4.4. Australia

The Royal Australian and New Zealand College of Radiologists published the “Ethical Principles for AI in Medicine” in April 2019. This document highlights the importance of upskilling medical practitioners and the development of standards and practices in the deployment of AI in Medicine and research [62].

The regulation of SaMDs falls under the Therapeutic Goods Administration (TGA). In August 2021, the Therapeutic Goods (Medical Devices) Regulation 2002 was amended, and a guideline entitled “Regulatory changes for software-based medical devices” was published to explain the amendments [63]. This guidance has been effective since February 2021 and includes a risk-based classification approach [63]. Areas exempt from the guidance are consumer health products for prevention and management; enabling technologies for telehealth, healthcare, and pharmaceutical dispensing; certain electronic medical records; population-based analytics; and laboratory information management systems [63]. This shows that the TGA has recognised the need for the regulation of AI-MDs at the national level, but it is also trying to harmonise at the international level by focusing on SaMDs with high-risk factors that have a high impact on patient safety.

### 4.5. China

The National Medical Products Administration (NMPA) of China, which provides regulatory oversight on medical products, published the “Technical Guideline on AI-aided Software” in June 2019. This guideline highlighted the characteristics of deep learning technology, controls for software data quality, valid algorithm generation, and methods to assess clinical risks. On 8 July 2021, the NMPA released the “Guidelines for the Classification and Definition of Artificial Intelligence-Based Software as a Medical Device”, which includes information on the classification and terminology of AI-MDs, the safety and effectiveness of AI algorithms, and whether AI-MDs provide assistance in decision making such as clinical diagnosis and the formulation of patient treatment plans [64]. On 7 March 2022, the Centre for Medical Device Evaluation under the NMPA published the “Guidelines for Registration and Review of Artificial Intelligence-Based Medical Devices”. These guidelines provide standards for the quality management of software and cybersecurity of medical devices taking into consideration the entire product’s lifecycle [64,65]. This shows that the NMPA has not only started to standardise the regulation of AI-MDs at the national level, but it is also trying to harmonise at the international level by focusing on risk factors and TPLC management through the publication of these guidelines [65].

### 4.6. Brazil

In September 2021, the Brazilian Legal Framework for AI, aiming to regulate the development and use of AI technology within Brazil, was approved by the Brazilian Chamber of Deputies. This legislation also prescribes a risk-based approach, but only for the development of AI solutions, and does not take into consideration the various applications of those AI solutions that may actually differ in terms of their risk [66]. Following this, the Brazilian Senate drew inspiration from regulatory plans for AI in OECD countries and received inputs from various stakeholders and the public to draft an AI law in December 2022. The draft AI law classifies health applications as high-risk AI systems, and hence they need to be maintained in a publicly accessible database that provides the details of the completed risk assessments of such systems [67]. The providers also need to conduct periodically repeated algorithmic impact assessments and must establish governance structures that facilitate the various rights of individuals. The rights to information, explanation, challenge, human intervention, non-discrimination, the correction of discriminatory bias, privacy, and the protection of personal data are included in those rights of individuals, irrespective of the risk classification of the AI system. The draft AI law also indicates that providers are strictly liable for any damages caused by their AI system [67].

### 4.7. Singapore

Singapore’s National AI Strategy aims to (i) identify areas to focus on and resources at the national level; (ii) set out how various stakeholders can work together to realise the positive impact of AI; and (iii) address areas where attention is needed to manage changes in and new forms of risk that arise when AI becomes more pervasive [68]. Further to this, on 25 May 2022, the Infocomm Media Development Authority (IMDA) launched the world’s first AI Governance Testing Framework and Toolkit called “AI Verify” for companies in Singapore that wish to demonstrate responsible AI in an objective and verifiable manner [69]. The testing framework consists of 11 AI ethics principles that jurisdictions around the world coalesce around and that are consistent with internationally recognised AI frameworks such as those from the EU, OECD, and Singapore’s Model AI Governance Framework. The 11 governance principles are transparency, explainability, repeatability/reproducibility, safety, security, robustness, fairness, data governance, accountability, human agency and oversight, inclusive growth, and societal and environmental well-being.

Specifically, with respect to the healthcare sector, the Health Sciences Authority (HSA) of Singapore released a second revision of its “Regulatory Guidelines for SaMD—A Lifecycle Approach” in April 2022, highlighting that the developers need to provide intended purpose, input data details, specifications of performance, control measures, and post-market monitoring and reporting. Good practices for AI developers and AI implementers are provided in “AI in Healthcare Guidelines” (AIHGIe), which was established by Singapore’s Ministry of Health (MOH) and published in October 2021. The recommendations in AIHGIe are based on the principles adapted from the AI Governance Framework established by the Personal Data Protection Commission (PDPC).

## 5. Limitations of This Study

The regulatory frameworks for AI in healthcare vary significantly across different jurisdictions, and this review may not have captured all the nuances and differences in the regulations. As this is a new and rapidly evolving area, there are very few studies that empirically evaluate the impact of specific regulations on AI in healthcare.

## 6. Conclusions

AI technologies have unmasked promising new solutions with tremendous potential to assist healthcare professionals in the navigation of the dynamic healthcare landscape with pressing healthcare challenges. In addition, there is also a need for regulatory frameworks and health system infrastructure to be agile as AI technologies evolve at a rapid pace. While most countries are regulating AI-MDs under traditional medical device software, provisions are being made to include good ML practices, holistic life cycle approaches, and strategies for the mitigation of risks associated with AI. The apps, software, and online platforms that can be combined with other health products or used standalone for healthcare purposes are also being included in regulatory frameworks. While most countries provide national guidelines for AI, the EU and Brazil aim to regulate it via a risk-based approach. Policy decisions are usually made based on clinical trials and published research outcomes. It is essential for regulators to factor in that published research may be biased towards positive outcomes or successful implementations of AI in healthcare. Negative experiences, challenges, or failures may be underreported, leading to a skewed perspective. The lack of standardised terminologies and definitions in the field of AI in healthcare may result in inconsistencies, thereby making it challenging to compare and synthesise information effectively.

The regulatory guidelines in AI have not yet converged, but there are ongoing discussions among the different governmental bodies. In June 2023, the US-EU Trade and Technology Council (TTC) came to a consensus for the development of a voluntary AI code of conduct, acting as an interim measure as the EU gears towards the passage of the AI Act. The US-EU TTC consensus established the building of trust and the fostering of cooperation in areas of technology governance and trade. The cornerstones of the draft voluntary AI code of conduct rely on transparency, risk auditing, and other technical details pertaining to the development of AI systems [70]. Fast-tracking the development of the US-EU AI code of conduct would be a step towards alignment between the US and Europe in AI legislation and policy. Thus, whether the global regulatory convergence would eventually occur remains unclear; however, it would be beneficial, given the shared healthcare challenges across countries and the country-agnostic nature of AI technologies.

## Figures and Tables

**Figure 1 healthcare-12-00562-f001:**
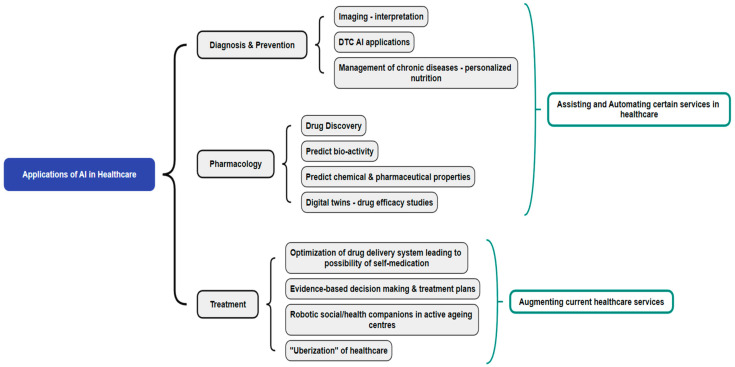
Applications of AI in healthcare services.

## Data Availability

The original contributions presented in this study are included in the article, further inquiries can be directed to the corresponding author.

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
