# Peer review of "Global Regulatory Frameworks for the Use of Artificial Intelligence (AI) in the Healthcare Services Sector"

_healthcare, 2024, doi:10.3390/healthcare12050562_

Round 1

Reviewer 1 Report

Comments and Suggestions for Authors

The  subject of the article  is  interesting as well as the adopted  methodology. I suggest a minor revision as follows:

Introduction: Please improve this part by adding the following references  that may help the reader to better understand the background.

-Lopreite, M., Panzarasa, P., Puliga, M. et al. Early warnings of COVID-19 outbreaks across Europe from social media. Sci Rep 11, 2147 (2021). https://doi.org/10.1038/s41598-021-81333-1

Methodology. The authors’ study focus on the literature review about the regulations pertaining to AI in the healthcare sector but in the section 1 and 2 they marginally discuss the state of art relative to this point. I suggest to describe better this point in the sections above.

Section 4: Please describe better the motivations to select only a group of countries for the “regulatory landscape”.

Please discuss the section  “Conclusion” in terms of “Policy implications”

Please discuss better the limitations of the study

A linguistic review is strongly suggested.

Comments on the Quality of English Language

A linguistic review is strongly suggested.

Author Response

Reviewer 1:

The subject of the article is interesting as well as the adopted methodology. I suggest a minor revision as follows:

Introduction: Please improve this part by adding the following references that may help the reader to better understand the background.

-Lopreite, M., Panzarasa, P., Puliga, M. et al. Early warnings ofCOVID-19 outbreaks across Europe from social media. SciRep 11, 2147 (2021). https://doi.org/10.1038/s41598-021-81333-1

The authors would like to thank the reviewer for the positive feedback and comments. The reference mentioned by the reviewer is on utilizing social media for identifying early warning signs of outbreaks and the authors feel that this reference may not be relevant to the introduction of the current manuscript as the current one is on the use of Artificial Intelligence. If we include the reference mentioned by the reviewer, it may be a forced inclusion and may not flow well with the topic at hand in the manuscript.

Methodology. The authors’ study focus on the literature review about the regulations pertaining to AI in the healthcare sector but in the section 1 and 2 they marginally discuss the state of art relative to this point. I suggest to describe better this point in the sections above.

Thank you to the reviewer for highlighting this and the search terms used for the study are now included in the methodology section from lines 212 to 215 (all line references are based on the clean version of the document).

Section 4: Please describe better the motivations to select only a group of countries for the “regulatory landscape”.

It is mentioned in the methodology section that the group of countries for the regulatory landscape were selected based on the reference provided as being the pioneers in the area of regulating AI in healthcare (reference number 42).

Please discuss the section “Conclusion” in terms of “Policy implications”

Policy implications have been included in the conclusion section now (lines 430 to 436).

Please discuss better the limitations of the study

The limitations of the study are also now provided prior to the conclusion section (lines 414 to 418).

A linguistic review is strongly suggested

A detailed linguistic review has been done by one of the authors which is shown in track changes in the document.

Reviewer 2 Report

Comments and Suggestions for Authors

The present paper seeks to provide an overview of the regulatory framework for the use of Artificial Intelligence in the health care sector. The authors conclude that it would be of benefit if “global regulatory convergence for AI in healthcare, similar to the voluntary AI Code of Conduct that is being developed by the US-EU Trade and Technology Council” to all nations.

Currently the paper has a number of shortcomings that preclude a recommendation for publication. First, there is no definition of artificial intelligence (AI). This is critical and would help the paper enormously, as in many cases the examples that the authors have selected should be classified as machine learning not true artificial intelligence. 

Second, many of the examples used by the authors to justify the adoption of AI are of limited real world relevance and indeed may add to costs as opposed to reducing them. Allied to this are the examples justifying AI many of which are potential solutions not proven examples. Indeed, one of the examples used by the authors for the UK, Babylon Health has seen the parent company filing for bankruptcy. 

Third, the authors have not provided the reader with any real details of their ‘literature search’. For example, what terms were used? What databases were explored? 

Fourth, there is a question over exactly what legislative agreements are necessary which would see all countries adopting a common legal framework. Currently, the authors have not provided the justification for their work. 

It is suggested that the authors rethink their paper, dropping the current introduction. The theme of their work is to provide the argument to address the first and fourth issues as the topic is of importance. They should also provide details of exactly how their ‘search’ was undertaken. 

I would also argue that currently their work is limited to healthcare. Any legal agreement covering AI must cover other sectors. How have those sectors responded to the challenges?

Comments on the Quality of English Language

Some changes to the manuscript would help its readability. However, the content is key and the above points must be addressed prior to any possible publication.

Author Response

Reviewer 2:

The present paper seeks to provide an overview of the regulatory framework for the use of Artificial Intelligence in the health care sector. The authors conclude that it would be of benefit if “global regulatory convergence for AI in healthcare, similar to the voluntary AI Code of Conduct that is being developed by the US-EU Trade and Technology Council” to all nations. Currently the paper has a number of shortcomings that preclude a recommendation for publication. First, there is no definition of artificial intelligence (AI). This is critical and would help the paper enormously, as in many cases the examples that the authors have selected should be classified as machine learning not true artificial intelligence.

The authors would like to thank the reviewer for the comments. The definition of AI is now provided in the introduction and its differentiation compared to machine learning is also provided (lines 43 to 49 - all line references are based on the clean version of the updated manuscript). Further, the authors would like to highlight that the examples cited mostly include a combination of machine learning and artificial intelligence. The authors have also added further examples to bring out the importance of AI (lines 123 to 125, 168 to 173, 181 to 185).

Second, many of the examples used by the authors to justify the adoption of AI are of limited real world relevance and indeed may add to costs as opposed to reducing them. Allied to this are the examples justifying AI many of which are potential solutions not proven examples. Indeed, one of the examples used by the authors for the UK, Babylon Health has seen the parent company filing for bankruptcy.

We would like to thank the reviewer for highlighting the point on Babylon Health which has now been removed even though the company has indicated in its website that it would soon find a taker to continue the process of ensuring safe and sound healthcare for its users. The authors have limited their examples to real-world relevance to showcase the potential of AI in healthcare as it would have maximum implications for regulatory frameworks. The authors have also added some points with regards to the cost implications before the methodology section (lines 193 to 201).

Third, the authors have not provided the reader with any real details of their ‘literature search’. For example, what terms were used? What databases were explored?

The government websites of the selected jurisdictions were included which is now mentioned and the search terms are also now included (lines 212 to 215).

Fourth, there is a question over exactly what legislative agreements are necessary which would see all countries adopting a common legal framework. Currently, the authors have not provided the justification for their work. It is suggested that the authors rethink their paper, dropping the current introduction. The theme of their work is to provide the argument to address the first and fourth issues as the topic is of importance. They should also provide details of exactly how their ‘search’ was undertaken.

The authors fully agree with the reviewer that it is necessary to see all countries adopting a common legal framework. However, as we have not yet reached that state, the authors have mentioned that different jurisdictions are trying to address this concept from different angles, and we have to wait to monitor how each of these jurisdictions are able to address the issues at hand with respect to AI in healthcare. The authors would like to highlight that it is important to monitor the existing state of governance in all these jurisdictions in order to reach convergence with regards to legislative agreements. The authors have also highlighted that while most countries are providing national guidelines for AI, some others are aiming to regulate it through a risk-based approach (lines 424 to 430). The authors have also indicated that currently discussions are ongoing between governmental bodies for convergence of regulatory frameworks as well (lines 438 to 443).

I would also argue that currently their work is limited to healthcare. Any legal agreement covering AI must cover other sectors. How have those sectors responded to the challenges?

Yes, the authors agree that this work is limited to healthcare and the main purpose of this work is to identify the crucial aspects and issues pertaining to use of AI in healthcare. The authors would like to highlight that the AI guidelines and laws are of general in nature, and they need to be applied to specific sectors. As the challenges and issues in other sectors may have different levels of risks and implications, the solutions to address such risks may not be applicable to healthcare which is a sector that involves the life of patients. Hence, the authors consider them as beyond the scope of this paper and this paper focuses only on issues specific to healthcare.

Reviewer 3 Report

Comments and Suggestions for Authors

This is very important isse due to healthcare sector is faced with challenges due to shrinking healthcare workforce and rise in chronic diseases that are worsening with the demographic and epidemiological shifts. The increasing use of AI presents many opportunities for health systems, but also many risks, such as the lack of human-to-human contact.

At the same time, it is necessary to note that AI-based technologies are still at a nascent phase and hence require various actors from AI researchers, developers to regulatory authorities to be informed of the developments of AI. Regulators should play a special role to ensure that there is adequate time for systems, medical personnel and patients to adapt to the legislation that will regulate AI.

The introduction is sufficient because it touches on the most important issues concerning AI, that is, the problems of the health services sector, the possibility of solving these problems with AI, the issue that we are learning AI and how to work with it all the time, and the relevance of regulations. The only thing the authors could point out and describe in a few sentences additionally is the cost to the systems associated with AI - whether it involves increased costs or reduced costs for the system. Or maybe both only in a certain order? Please explain.

The material and methods section does not describe how the authors conducted their literature review. There are no assumptions for this review, no guidelines, and no criteria for what publications were considered for review. This section needs to be completed.

Breaking down and describing the legislative guidelines by country is a very interesting idea, but I miss the comparison of these guidelines - perhaps with a table or in some other graphic way?

The literature review does not require a discussion of the results, while it would be good to juxtapose what has been obtained with other reviews on the topic - if any, since I know that the topic of AI in health care is quite new.

References is prepared carefully, there are minor typos - please check it again carefully.

Author Response

Reviewer 3:

This is very important issue due to healthcare sector is faced with challenges due to shrinking healthcare workforce and rise in chronic diseases that are worsening with the demographic and epidemiological shifts. The increasing use of AI presents many opportunities for health systems, but also many risks, such as the lack of human-to-human contact. At the same time, it is necessary to note that AI-based technologies are still at a nascent phase and hence require various actors from AI researchers, developers to regulatory authorities to be informed of the developments of AI. Regulators should play a special role to ensure that there is adequate time for systems, medical personnel and patients to adapt to the legislation that will regulate AI. The introduction is sufficient because it touches on the most important issues concerning AI, that is, the problems of the health services sector, the possibility of solving these problems with AI, the issue that we are learning AI and how to work with it all the time, and the relevance of regulations. The only thing the authors could point out and describe in a few sentences additionally is the cost to the systems associated with AI -whether it involves increased costs or reduced costs for the system. Or maybe both only in a certain order? Please explain.

The authors would like to thank the reviewer for the positive feedback. The implication to cost has now been added to the manuscript before the methodology section (lines 193 to 201).

The material and methods section does not describe how the authors conducted their literature review. There are no assumptions for this review, no guidelines, and no criteria for what publications were considered for review. This section needs to be completed.

The details of the methods, that is the search terms and resources that were included are now added in the methodology section (lines 212 to 215).

Breaking down and describing the legislative guidelines by country is a very interesting idea, but I miss the comparison of these guidelines - perhaps with a table or in some other graphic way?

The comparison of these guidelines is provided in the conclusion section where it is highlighted that there are mainly two approaches used in the selected jurisdictions – a soft law approach of using guidelines and a risk-based regulatory approach (lines 424 to 430).

The literature review does not require a discussion of the results, while it would be good to juxtapose what has been obtained with other reviews on the topic - if any, since I know that the topic of AI in health care is quite new.

Yes, as the reviewer has correctly identified, this is a new area and this study is the first of its kind to review the regulatory frameworks from a global perspective. A paragraph on limitations is added before the conclusion section to highlight this and to address the reviewer’s comment with regards to the other reviews on this topic (lines 414 to 418).

References is prepared carefully, there are minor typos – please check it again carefully.

The authors use “Zotero” software with in-built style repository for the “Healthcare” journal and they assure that all the references have now been checked for typos. Thank you!

Round 2

Reviewer 2 Report

Comments and Suggestions for Authors

The present paper is a resubmission and the authors have undertaken a substantial revision of their work. The paper is now of a far higher quality than the previous version  and the authors have added a number of supportive references which aids the readability. There are two minor queries I would raise that the authors may like to consider addressing.

First, the authors have highlighted a number of jurisdictions in which AI is being explored. A paragraph bringing together commonalities and differences overall would add the value of the paper.

Second, many of the economic arguments presented are I would argue hypothetical. The authors may wish to consider rewording certain elements the term ‘reallocating’ is I think more appropriate as this allows more patients or treatments to be provided for the same resources. 

Author Response

Reviewer 2: second round of comments:

The present paper is a resubmission and the authors have undertaken a substantial revision of their work. The paper is now of a far higher quality than the previous version and the authors have added a number of supportive references which aids the readability. There are two minor queries I would raise that the authors may like to consider addressing.

The authors would like to thank the reviewer for reviewing the manuscript again and also for providing positive feedback.  

First, the authors have highlighted a number of jurisdictions in which AI is being explored. A paragraph bringing together commonalities and differences overall would add the value of the paper.

 Yes, this is provided in the conclusion section from lines 484 to 490 in the updated version as follows: While most of the countries are regulating AI-MDs under traditional medical device software, provisions are being made to include good ML practices, holistic life cycle approaches, and mitigation of risks associated with AI. The apps, software and online platforms that can be combined with other health products or used standalone for healthcare purposes are also being included in regulatory frameworks. While most of the countries are providing national guidelines for AI, EU and Brazil are aiming to regulate it via a risk-based approach.”

Second, many of the economic arguments presented are I would argue hypothetical. The authors may wish to consider rewording certain elements the term ‘reallocating’ is I think more appropriate as this allows more patients or treatments to be provided for the same resources. 

Yes, the authors agree that they are hypothetical as they are based on the existing literature review and they have now reworded it accordingly in line 233.